# Multi-output prediction of global vegetation distribution with incomplete data

**Rita Beigaite**[1]  **Jesse Read**[2]  **Indre Zliobaite**[1]

## Abstract

As the climate is changing, large changes in vegetation distribution are already taking place and expected to happen in the future. Our goal is to explore the links between climate and vegetation, and build a predictive model mapping climatic conditions to vegetation cover, based on global remote sensing data. The main challenge is that many areas in the world are already significantly impacted by human activities, and natural mosaic of vegetation is altered, which makes natural vegetation data incomplete and increasingly inaccurate after fractions of agricultural and urban activity are removed. Here we employ multi-output feed-forward neural networks for predicting natural vegetation cover from local climatic conditions. We conduct experiments to evaluate how accurate predictions of the vegetation fraction can be and how they are affected by human-altered observations. Results show that it is possible to make such predictions with high accuracy even if the training data are incomplete.

## 1. Introduction

Similar climatic conditions create environments where similar vegetation types can exist (Adams, 2009). Good understanding of such links can help to model and predict future changes in vegetation distribution. In this study, we aim to predict the fractions of natural land cover type for given areas of land from local climatic conditions. The natural land cover types here are the ones which are determined only by climatic conditions and exist independently of or without human intervention (Figure 1). The data we are analyzing are remote sensing data. Each observation holds values of climate variables and corresponding fractions of land cover types. The main challenge of this prediction task

[1]Dept. of Computer science, University of Helsinki, Finland [2]LIX, CNRS, Ecole Polytechnique, Institut Polytechnique de Paris, France. Correspondence to: Rita Beigaite <rita.beigaite@helsinki.fi>.

*Presented at the first Workshop on the Art of Learning with Missing Values (Artemiss) hosted by the $37^{th}$ International Conference on Machine Learning (ICML). Copyright 2020 by the author(s).*

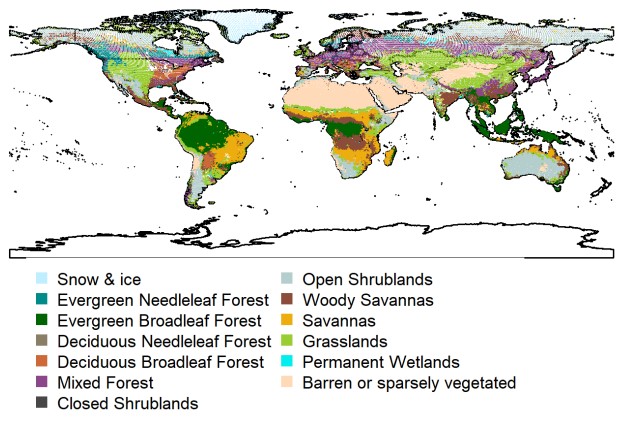

Figure 1. Distribution of natural land cover by dominant type

is that many parts of the landscape are altered by excessive human activities and changed into urban areas or croplands, and it is unknown which natural vegetation types have been replaced by human activity land cover types, and to what extent. Therefore, part of the information on what would be the real distribution of land vegetation, given current climatic conditions, is not available. For example, in one observation 5% of land is urban area, 15% grassland, 50% forest and 30% shrublands. When we discard human activity fraction, we are left with 95% covered grid cell. It is incomplete as we do not know how those 5% would have been distributed. For example, these 5% could have been only forest or maybe 2.5% was forest and 2.5% was grasslands, or 1% forest, 1% grassland, 3% shrublands and so on. We seek to find the global patterns with the intention to better understand what would be the distribution of the vegetation if the landscape would have not been excessively altered by humans. We approach this in two main tasks. Firstly, we need to predict the fraction of each vegetation type in relation with other vegetation types. This should be done under the constraint that all fractions have to sum up unity. One way to address the fraction prediction problem is to look at it as a compositional data analysis problem (Aitchison, 1982; Pawlowsky-Glahn & Buccianti, 2011). Such data are being analyzed in various fields: demography (Lloyd et al., 2012), economics (Ferrer-Rosell et al., 2015), chemistry and geology (Buchanan et al., 2012). For example, in (Buchanan

et al., 2012) authors describe high resolution prediction of the soil particle-size fractions and highlight that most commonly applied methods as multiple linear regression does not follow the requirements (non-negativity and summation to unity) of compositional data. Therefore, the authors of the paper transform the data from simplex to real space by using additive log-ratio transformation, which is widely used for this purpose. However, log-ratio transformations fail in case of presence of zero values in the composition (Wang et al., 2007). Our vegetation fraction data has many sites where one or more vegetation type is absent. Thus, if we would use compositional data analysis approach, zero values would require specific treatment. Another way is to define this problem as a multi-output (multi-target) regression problem (Borchani et al., 2015) which aims to simultaneously predict multiple real-valued output/target variables. In (Kocev et al., 2009) a multi-target regression tree was used for prediction of the quality of the remnant indigenous vegetation across an extensive area of south-eastern Australia. The authors state that one of the main advantages of multi-target prediction over separate models for each target is that multi-output models can capture underlying relationships between the targets. It is an important advantage in case of the fraction data as its compositional structure make the targets dependant on each other. In addition, the multi-target models have a smaller size and thereby are faster to learn and apply. In (Li & Yeh, 2002) a three-layer neural network was employed for the simulation of multiple land uses. The authors explain that, as many geographical phenomena variables are correlated, the neural networks provide an advantage over traditional regression models by not necessarily requiring for spatial variables to be independent of one another, as these dependencies are modeled by hidden layers. Secondly, since only around 20% of all observations are in the areas minimally impacted, or not impacted, by humans, we need to infer the fraction of vegetation types reduced by human activity or replaced by water, i.e., we need to deal with incomplete data. In this article, the term *incomplete data* is not used as an analogous term for the missing data problem. The missing data problem is defined as lack of information for some variables for some cases (Allison, 2001). However the values of the vegetation fractions are not completely missing. In most of the cases the fractions of the vegetation are only reduced by unknown amount and does not sum up exactly to unity. In around 1% of all observations in our data the land cover is composed solely of human activity types or/and water. Only in such cases the values of natural vegetation fractions are completely missing. For the issue of incomplete data, if approached as a missing data problem, we note that several different approaches could be applied. In (Little & Rubin, 2019) these approaches are broadly grouped into four categories: procedures based on completely recorded units, weighting procedures, imputation, model based methods. However, the incompleteness

of our vegetation fraction data resembles weakly supervision learning problem (Zhou, 2017), common, for instance in computer vision tasks. Although, it has not been extensively analyzed for regression problems. We design and experiment with neural networks architectures to evaluate how accurate predictions of the vegetation types can be, as well as how much are they affected by incomplete observations. Results show that with our approach it is possible to achieve high prediction accuracy. Additionally, we analyze which fractions of different types of vegetation are difficult to predict or not predicted as accurately as others.

## 2. Problem Definition

We have a matrix

$$X \in \mathbb{R}^{52297 \times 47}$$

where each row corresponds to 47 climatic features for a given area of land; 52297 areas in total.

The aim is to produce an output matrix

$$\hat{Y} \in [0, 1]^{52297 \times 13}$$

under the constraint that each row $[\hat{y}_{i,1}, \ldots, \hat{y}_{i,13}]$ represents a distribution over vegetation types (listed in Figure 1 and Table 2); i.e., under the constraint $\sum_j \hat{y}_{i,j} = 1$ so as to be a valid distribution. However, the output matrix which we use for training is:

$$\tilde{Y} \in [0, 1]^{52297 \times 13}$$

under the assumption that

$$\tilde{y}_{i,1} + \tilde{y}_{i,2} + \cdots + \tilde{y}_{i,13} + h_i + w_i = 1 \qquad (1)$$

where

$$h_i = h_{i,1} + h_{i,2} + \cdots + h_{i,13}, \quad h \in [0, 1], \text{ and}$$

$$w_i = w_{i,1} + w_{i,2} + \cdots + w_{i,13}, \quad w \in [0, 1]$$

represent the fractions of human impact and water bodies, respectively, in the $i$-th region.

On top of the constraint of ensuring a valid distribution that sums to unity for each row of output, the main challenge is that it is not known how $h$ and $w$ are distributed over different vegetation types, hence the incomplete data problem.

## 3. Data

Our data set consists of climatic variables and corresponding vegetation fraction variables. Climatic variables are from two open access data sources: BIOCLIM data (Fick & Hijmans, 2017) and Climdex Climate extreme indices (CEI) (Sillmann et al., 2013a;b), as well as the Precipitation

minus Potential Evaporation (PET) variable derived from the cru_ts4.01 dataset (Mitchell & Jones, 2005; Harris et al., 2014). PET variable was added to our data set for capturing the aridity of the environment. BIOCLIM data set is made of observations averaged over thirty years (1970-2000) and includes annual averages, ranges or max-min values, for example, mean annual temperature, mean annual rainfall or max temperature of warmest month. CEI data set consists of observations which are averaged over a twenty-year period (1979-2010) and includes various climate extremes such as maximum length of dry spell (consecutive days without a rainfall), monthly minimum value of daily maximum temperature. The vegetation fraction variables are from MODIS (Channan et al., 2014) land cover product (MCD12C1), year 2001. BIOCLIM, CEI and MODIS data as well as PET variable were interpolated onto a 10min × 10min grid then nearest interpolation onto 50km × 50km grid.

## 4. The Proposed Approach

For solving the vegetation fraction prediction problem we employ feed-forward multi-output neural networks. We perform the analysis from two perspectives: training the neural network for evaluation of prediction accuracy and learning how different strategies of dealing with incomplete data affects these prediction results.

### 4.1. Baseline Methods and Neural Network

To compare the results of the neural network, we selected three trivial solutions as baselines. The first approach is equivalent to the *majority class* concept. We predict each fraction value to be equal to one for the land cover type, which occupies the biggest territory worldwide, and zero for the other types. The second approach predicts fractions of all types to be equal. That is, all vegetation types are predicted to occupy approximately 0.077 fraction of the grid cell. The third approach predicts the fractions to be of the same size as the distribution of each type in the training data. For example, *Grasslands* are predicted to occupy approximately 0.16 part of the grid cell.

The proposed feed-forward neural network comprises of three layers. The first layer consists of 47 input neurons for each climatic feature. The hidden layer has 20 neurons and is activated by the Rectified Linear Unit (ReLU) activation function. The third layer consists of 13 output neurons for each vegetation type. In order to satisfy the constraint of the outputs summing up to unity, we apply the *softmax* function in the output layer. The architecture of the neural network was chosen based on the values of ten-fold cross-validation errors (root mean squared error, mean absolute error) of the initial experiments. The neural networks were trained using the *mean absolute error* loss function, and *RMSProp* optimizer. All experiments were carried out using the *Keras*

(Chollet, 2015) library. Computing resources were provided by the Finnish Grid and Cloud Infrastructure (persistent identifier urn:nbn:fi:research-infras-2016072533).

### 4.2. Incomplete Data Approaches

We tested several approaches to deal with incomplete data:

1. Basic approach of the missing data. That is, discarding incomplete observations and conducting analysis only on complete data.
2. Re-scaling each observation to sum up to unity.
3. Imputation based on latitudes. For each latitude, we calculate averages of the vegetation fractions in complete observations. Instead of substituting incomplete observations of the same latitude with these averages, we only fill in the missing parts. The difference between averaged observation and incomplete observation is used as proportion guideline of how much of each vegetation type should be filled in.
4. Imputation based on latitudes and elevation. The difference from the previous approach is that we calculate averages for each combination of latitude and elevation. Elevation values are divided into 5 range categories.
5. Using incomplete data without alterations together with complete observations (all observations).
6. Using asymmetric Loss Function when training neural networks. We expect the neural network to predict the true fractions which would be either of the same or higher value than in incomplete observation. Therefore, we consider asymmetric loss function described in (Elliott et al., 2005). We penalize MAE loss multiplying by real number $p = 2$ in case of under-prediction when predicted value is smaller than the ground-true value.

### 4.3. Performance Evaluation

The results of the experiments are compared using root mean squared (RMSE) and mean absolute (MAE) errors. We evaluate the performance of all the fraction prediction experiments only on complete (C) observations as these errors would be misleading computed on incomplete (INC) data when the outputs are trained to sum up to unity. However, the complete observations are not uniformly distributed across the world. For evaluating models performance on both complete and incomplete data, we compare how well the dominant vegetation type is predicted and measure the prediction accuracy as in classification task. The dominant vegetation type is considered to be the one which occupies the largest fraction in a grid cell. In this approach, we consider only those observations where dominant type remains the same even if the human activity fraction would be added to the second largest vegetation type. In this case, we are able to evaluate predictions on around 80% of all observations.

*Table 1.* Prediction errors ($\times 10^2$) on the test set and accuracy for the dominant type

| APPROACH | TRAINING DATA | MAE | RMSE | ACC. (C / INC) |
|---|---|---|---|---|
| NEURAL NETS | ONLY COMPLETE OBSERVATIONS | 1.36 | 3.20 | 94% / 63% |
| | RE-SCALED OBSERVATIONS | 2.14 | 5.07 | 88% / 80% |
| | MISSING PARTS IMPUTED (LATITUDE) | 2.16 | 5.19 | 93% / 80% |
| | MISSING PARTS IMPUTED (LATITUDE & ELEVATION) | 1.30 | 3.01 | 93% / 80% |
| | ALL OBSERVATIONS | 2.12 | 4.94 | 93% / 80% |
| NEURAL NETS (ASYMM. LOSS) | ONLY COMPLETE OBSERVATIONS | 1.39 | 3.23 | 93% / **63%** |
| NEURAL NETS (ASYMM. LOSS) | ALL OBSERVATIONS | **1.25** | **2.89** | 93% / 80% |
| BIGGEST TYPE | - | 8.0 | 18.1 | - |
| EQUAL PROPORTION | - | 13.8 | 19.6 | - |
| DISTRIBUTION OF DATA | - | 11.5 | 17.0 | - |

*Table 2.* Prediction errors ($\times 10^2$) on the test set

| VEGETATION TYPE | MAE | RMSE |
|---|---|---|
| EVERGREEN NEEDLELEAF FORESTS | 0.30 | 3.38 |
| EVERGREEN BROADLEAF FORESTS | 0.51 | 4.91 |
| DECIDUOUS NEEDLELEAF FORESTS | 0.14 | 1.32 |
| DECIDUOUS BROADLEAF FORESTS | 0.13 | 3.10 |
| MIXED FORESTS | 0.29 | 2.61 |
| CLOSED SHRUBLANDS | 0.18 | 1.87 |
| OPEN SHRUBLANDS | 4.59 | 12.69 |
| WOODY SAVANNAS | 1.33 | 5.89 |
| SAVANNAS | 1.10 | 5.06 |
| GRASSLANDS | 4.02 | 10.93 |
| PERMANENT WETLANDS | 0.11 | 1.01 |
| PERMANENT SNOW & ICE | 0.35 | 3.31 |
| BARREN OR SPARSELY VEGETATED | 3.25 | 11.16 |

## 5. Results and Implications

The predictions of our neural network yields lower errors than chosen baseline approaches (Table 1). Therefore, we consider our model to be reasonably informative. The experiments with different incomplete data approaches show that the most accurate results can be achieved by using asymmetric loss function in model training when both complete and incomplete observations are used. Prediction errors of using only complete observations in model training are one of the lowest. However, experimental results on evaluating the prediction accuracy of dominant vegetation types (Table 1) show that using this approach the prediction accuracy of incomplete observations is only 62%. This suggests that complete observations does not carry full information about distribution of the natural vegetation worldwide as removing incomplete observations leads to the loss of majority of data points in Europe, North America as well as India. Imputation based on only latitude and re-scaling approaches did not lead to any improvement in prediction errors. However, combining elevation with latitude seems to yield similar to asymmetric loss function approach results. Values in Ta-

ble 2 represent the mean errors of different predicted types of land cover. If we analyze the errors of each vegetation type separately, it is clear that not all fractions of land cover types can be predicted equally well. For instance, fractions of *Grasslands*, *Open Shrublands* and *Sparsely vegetated* land covers are predicted with at least four times higher error than any other land cover type. One of the possible reasons for this could be that these land cover types can exist in very similar or, in some cases, the same climatic conditions.

## 6. Conclusions

We analyzed vegetation type from climatic conditions, and employed a neural network architecture suitable for predicting the composition of vegetation cover in the presence of incomplete data. Our experimental results show that we can predict vegetation fractions with high accuracy. However, for some of the vegetation types prediction error is higher than for others. Those types can coexist in very similar climatic conditions, and thus, their proportions can be easily mixed by the model. The results also indicate that by using asymmetric loss function or imputing incomplete data based on latitude and elevation, it is possible to train the model on both complete and incomplete data at the same time without increasing prediction error of complete observations. When we include incomplete data into the training set, different parts of the world are more equally represented and the model is not adapted only to complete observations. In this way, we can model potentially more accurate worldwide links between vegetation and climate.

## 7. Acknowledgments

We thank Hui Tang for initial pre-processing of the data.

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
