# OpenReview forum: "Multi-output prediction of global vegetation distribution with incomplete data"
_ICML.cc/2020/Workshop/Artemiss — ICML Artemiss 2020_

### Official Review · AnonReviewer1 · 2020-06-20
**Missing data problem in vegetation type prediction based on climatic variables using feedforward net.**

**Rating:** 6
**Confidence:** 3

**Review:**

Summary:
A feed-forward neural network based model is used to predict vegetation cover from climatic conditions, when the landscape is altered due to human activities resulting in "missing data". The authors make it clear that this is more like a weakly supervised task than an incomplete data problem. They compare the model with simple baselines and show the neural networks when using an asymmetric loss performs better at the prediction task.

Strengths:
+ The problem is well motivated and the distinction to classical incomplete data problems is clear.
+ Although simple, the experiments are convincing

Weaknesses:
- Table 1 is difficult to parse as the bulleted list in Sec. 4.2 and the entries in the table do not exactly correspond to each other. For instance, it is unclear what "all observations" are? Which imputation is performed to go from incomplete to all observations?
- Fig.1: What is the source for the data shown in it?
- Several claims about the proportion of vegetation, in the Introduction are not backed up by citations. Are these hypothetical numbers or based on actual data?

Minor comments:
- Sec. 1 is all in a single paragraph. Would urge the authors to structure it. Something on the lines of : Intro->Problem->Related work->Contributions, or something like that.

---

### Official Review · AnonReviewer2 · 2020-06-23
**Multi-output feed-forward networks for the prediction of vegetation type distribution with incomplete data**

**Confidence:** 4
**Rating:** 6

**Review:**

The authors use feed-forward neural networks to predict the distribution of vegetation cover in areas described by the climatic features. Due to the human impact, some parts of these areas cannot be assigned to any of the 13 vegetation types. This issue is formulated as an incomplete data problem, which is similar in some sense to the missing data problem. The authors present several solutions, including the imputation of obscured values, exploiting the distribution of the remaining part of the area, and using the asymmetric loss function that was introduced to correct under-predictions. In the experimental section, these approaches are compared with simple baselines. It was also demonstrated, that the effective use of incomplete data leads to significantly better results.

Pros:
- The script is clearly written, and its topic is relevant to the missing data problem.
- The proposed solutions, especially the one involving the asymmetric loss function, give good results for the presented task.

Cons:
- Both Tables could be described better, e.g. which model was used in Table 2 and what INC in Table 1 stands for - is this calculated only for the incomplete data? If so, why accuracy for the full data (C+INC) is not reported?
- It is difficult to interpret MAE and RMSE when comparing distributions. A metric that measures similarities between distributions should also be considered, e.g. the chi-squared distance.
- Information about the data splitting method should be provided. The lack of the description on how the testing set was created, combined with the low prediction errors, raises the question whether, e.g., areas that are close to each other and have very similar vegetation type distributions could be placed in both the training and testing sets.

---

### Decision · Program_Chairs · 2020-07-02

**Decision:**

Accept

**Comment:**

We're happy to accept this paper at Artemiss. We'll contact you soon to inform you about more details concerning the format of your presentation at the workshop, and the camera-ready version deadline. Please take into account the referee's comments to write the camera-ready version.